# Atomic Scale Optimization Strategy of Al-Based Layered Double Hydroxide for Alkali Stability and Supercapacitors

**DOI:** 10.3390/ijms231911645

**Published:** 2022-10-01

**Authors:** Chuan Jing, Kai Shu, Qing Sun, Jiayu Zheng, Shuijie Zhang, Xin Liu, Kexin Yao, Xianju Zhou, Xiaoying Liu

**Affiliations:** 1College of Science, Chongqing University of Posts and Telecommunications, Chongqing 400065, China; 2School of Chemistry and Chemical Engineering, Chongqing University, Chongqing 400044, China; 3Engineering Research Center for Waste Oil Recovery Technology and Equipment of Ministry of Education, College of Environment and Resources, Chongqing Technology and Business University, Chongqing 400067, China

**Keywords:** layered double hydroxide, alkali etching, alkali stability, electrochemical performance, supercapacitor

## Abstract

The pseudocapacitor material is easily decomposed when immersed in alkaline solution for a long time. Hence, it is necessary to find a strategy to improve the alkali stability of pseudocapacitor materials. In addition, the relationship between alkali stability and electrochemical performance is still unclear. In this work, a series of Al-based LDH (Layered double hydroxide) and derived Ni/Co-based sulfides are prepared, and corresponding alkali stability and electrochemical performance are analyzed. The alkali stability of CoAl LDH is so poor and can be improved effectively by doping of Ni. Ni_1_Co_2_S_4_ and Ni_2_Co_1_Al LDH exhibit an outstanding alkali stability, and Ni_2_Co_1_S_4_ exhibits an extremely poor alkali stability. The variable valence state of Co element and the solubility of Al in alkali solution are the fundamental reasons for the poor alkali stability of CoAl LDH and Ni_2_Co_1_S_4_. Ni_2_Co_1_S_4_ showed an outstanding electrochemical performance in a three-electrode system, which is better than that of Ni_1_Co_2_S_4_, indicating that there is no direct correlation between alkali stability and electrochemical properties. Sulfidation improved the electrical conductivity and electrochemical activity of electrode materials, whereas alkali etching suppressed the occurrence of the electrochemical reaction. Overall, this work provides a clear perspective to understand the relationship between alkali stability and electrochemical properties.

## 1. Introduction

Supercapacitors are a new generation of energy storage devices with the characteristics of long cycle life, safety, environmental friendliness, and outstanding power density [1,2]. To make supercapacitors more efficient, two main types of supercapacitors have been developed, such as electric double-layer capacitors and pseudocapacitors, in which pseudocapacitors store energy through reversible redox reactions primarily [3]. Up to now, different kinds of transition metal hydroxide (NiCo LDH [4], NiMn LDH [5], etc.), oxide (Fe_2_O_3_ [6], MoO_3_ [7], etc.), sulfide (CuCo_2_S_4_ [8], Ni_2_Co_1_S_4_ [9], etc.), and phosphide (FeP_4_ [10], NiCoP [11], etc.) are used as the pseudocapacitor materials due to the ultrahigh theoretical specific capacitance and abundant active sites. The main application environment of pseudocapacitor materials is alkaline electrolyte, such as KOH and NaOH. Prolonged immersion in alkali solution may cause problems of material decomposition.

Numerous studies are shown that CoAl LDH possesses extremely poor alkali stability and is easily decomposed into CoOOH or Co(OH)_2_ in alkali solution due to the alkali instability of Al. For example, Lu et al. [12] reported that porous CoAl LDH are obtained when immersed CoAl LDH in 5 M NaOH for 5 min due to the dissolution of Al element. Xu et al. [13] studied the crystal phase transformation of CoAl LDH with alkali etching time-variation. The results showed that CoAl LDH are converted to Co(OH)_2_ and CoOOH gradually, and the structure becomes porous. Similar behavior can be found in Zn-based hydroxides. For example, Zhao et al. [14] used ZnCr LDH as the precursor to prepare porous material with abundant oxygen vacancy at 1 M NaOH solution. He et al. [15] removed Zn from ZnNiCo LDH completely by alkali etching at 2 M NaOH. Based on the above operation, abundant nickel vacancies are obtained, and the application performance is improved.

Despite the Al-based or Zn-based hydroxides, phosphides and oxides also exhibit an unstable property in alkali solution. For example, Li et al. [16] used 6 M KOH to etch the CoP for 24 h. The results showed that CoP decomposed into CoOOH and Co(OH)_2_ completely. Similarity, NiMgAl oxide also can be etched by NaOH due to the unstable element of Al. Yong et al. [17] used NiMgAl oxide as the precursor to prepare porous catalyst support at 0.5 M KOH. In addition, Li et al. [18] used 10 M KOH to treat NiCo sulfides for 3 h. The obtained material exhibits an abundant porous structure. Hao et al. [19] studied the phase transformation behavior of Bi_2_MoO_4_ at 0.1 M NaOH. Some of Bi_2_MoO_4_ decomposed to the α-Bi_2_O_3_ gradually. Liu et al. [20] prepared CoFe porous nanowires by the alkali etching of CoFe oxide. The obtained material provides abundant active sites for subsequent application. The above cases show that, many electrode materials, especially Al-containing materials, have the problem of alkali instability. Hence, it is necessary to find a suitable method to improve the alkali stability of these electrode materials.

Interestingly, our recent study (Appendix A) found that NiAl LDH has superior alkali stability. The crystal phase of NiAl LDH is not destroyed by the prolonged immersion in alkali solution. Doping the element of Ni in CoAl LDH may improve the alkali stability of CoAl LDH. It is worth mentioning that the electrical conductivity of hydroxide is poor when compared sulfide. Zhou et al. [21] fabricated NiV-S by the sulfidation of NiV-LDH, and the obtained NiV-S presents a much higher specific capacitance when compared with the pristine NiV-LDH. Qin et al. [22] prepared NiCo_2_S_4_ from sulfidation of NiCo-LDH. After complete sulfidation, the specific capacitance, rate capacity, and cycling stability are improved obviously. Hence, the conversion of hydroxide to sulfide is an effective strategy to improve the electrochemical performance of electrode material. Upon further analysis, is there a relationship between electrical conductivity and alkali stability? Is there a positive correlation between alkali stability and electrochemical properties? The above questions now are not clear.

In this work, a class of Al-based LDH and derivate Ni/Co-based sulfides is prepared by combining the hydrothermal process and sulfidation. The alkali stability and electrochemical performance of as-prepared materials are analyzed clearly. The alkali stability of CoAl LDH is so poor and can be improved effectively by doping of Ni. Ni_1_Co_2_S_4_ and Ni_2_Co_1_Al LDH exhibit an outstanding alkali stability, and Ni_2_Co_1_S_4_ exhibits an extremely poor alkali stability. The variable valence state of Co and the solubility of Al in alkali solution may be the fundamental reasons for the poor alkali stability of CoAl LDH and Ni_2_Co_1_S_4_. Ni_2_Co_1_S_4_ showed excellent cycling stability of 85.48% capacitance retention after 5000 cycles in the three-electrode system, which is better than that of Ni_1_Co_2_S_4_ obviously, indicating that there is no direct correlation between alkali stability and electrochemical properties. In addition, alkali etching and sulfidation have opposite effects on the electrochemical properties of electrode materials, the former is negative, the latter is positive. Overall, this work provides us with a clear perspective to understand the relationship between alkali stability and electrochemical properties.

## 2. Results

### 2.1. Morphology and Structural Characteristics

Appendix A present the XRD patterns of NiAl LDH and CoAl LDH at different concentrations of KOH. NiAl LDH presents a remarkable alkali stability. Alkali etching leads to the slight decrease of crystallinity of NiAl LDH without any change of crystalline. However, CoAl LDH presents a poor alkali stability. The crystalline phase of CoAl LDH is firstly converted to CoOOH and then to Co(OH)_2_. As is well-known, the alkaline environment is necessary for LDH-based electrode materials to produce electrochemical performance. Hence, alkali stability may be an important factor affecting the electrochemical stability of electrode materials. According to the Appendix A, the CoAl LDH-E obtained by alkali etching at 3 M KOH presents a terrible specific capacitance of only 85.35 F g^−1^ at 1 A g^−1^. For comparison, CoAl LDH presents an outstanding specific capacitance of 838.03 F g^−1^ at 1 A g^−1^. Based on the outstanding alkali stability of NiAl LDH, prepared NiCoAl LDH may be a preferable method to improve the alkali stability of CoAl LDH. Hence, the Ni_1_Co_2_Al LDH and Ni_2_Co_1_Al LDH are prepared. Considering the positive influence of electrical conductivity, the derived Ni_1_Co_2_S_4_ and Ni_2_Co_1_S_4_ from Ni_1_Co_2_Al LDH and Ni_2_Co1Al LDH are also prepared. The schematic illumination of the synthesis and alkali etching of NiCo sulfides is shown in Figure 1.

The XRD patterns of the as-prepared samples were shown in Figure 1a–c. As shown in Figure 1a, all the XRD patterns exhibit a standard LDH characteristic peaks of (003) and (006) planes at around 11° and 23°, respectively. From Figure 1b, the interlayer spacing of the as-prepared samples are found in the order as Ni_1_Co_2_Al LDH < Ni_2_Co_1_Al LDH = Ni_1_Co_2_Al LDH-E < Ni_2_Co_1_Al LDH-E due to the content variation of a trivalent ion (Co^3+^ and Al^3+^). For example, the interlayer spacing of Ni_1_Co_2_Al LDH is 7.654 Å. The interlayer spacing of the obtained Ni_2_Co_1_Al LDH is increasing to 7.694 Å with the increase of the content of Co^3+^ due to the receding charge density of host layer caused by the decrease of the Co^3+^. The interlayer spacing of Ni_2_Co_1_Al LDH-E and Ni_1_Co_2_Al LDH-E are increased to 7.748 and 7.694 Å, respectively, due to the receding charge density of host layer caused by the decrease of the leaching of Al^3+^ after the alkali etching process. From red and green lines of Figure 1c, the peaks at around 31.58°, 38.32°, and 55.31° are assigned to the (311), (400), and (440) planes of cubic Ni_1_Co_2_S4 (JCPDS No. 73–1704), suggesting the success of preparation process of Ni_1_Co_2_S_4_ and Ni_2_Co_1_S_4_. From the yellow line, the peaks at 20.24°, 34.41°, 38.89°, 59.98°, 61.25°, and 65.34° are assigned to the (003), (012), and (110) planes of hexagonal CoOOH (JCPDS No. 07-0169) and the (012), (110), and (113) planes of Ni(OH)_2_·0.75H_2_O (JCPDS No. 38-0715). The result suggests that Ni_2_Co_1_S_4_ have been decomposed into CoOOH and Ni(OH)_2_ due to its alkali instability. For comparison, the XRD pattern of Ni_1_Co_2_S_4_ has almost no change due to the superior alkali stability. As shown in Figure 1d, obvious peaks at 1352 and 576 cm^−1^ in spectra of Ni_2_Co_1_Al LDH, Ni_2_Co_1_Al LDH-E, and Ni_2_Co_1_S_4_-E are assigned to the characteristic adsorption of CO_3_^2−^ and M-O (Ni-O and Co-O) components, respectively, indicating that the Ni_2_Co_1_S_4_-E has been decomposed into hydroxide. The broad peak at around 536 cm^−1^ is assigned to the characteristic adsorption of M-S (Ni-S and Co-S).

From Figure 2a, Ni, Co, and S can be found in XPS spectra of Ni_2_Co_1_S_4_ and Ni_2_Co_1_S_4_-E, and the Ni, Co, Al O, and C can be found in Ni_2_Co_1_Al LDH and Ni_2_Co_1_Al LDH-E, the chemical composition of these as-obtained samples is in line with forecast, indicating that success of the preparation process. From Figure 2b,d, the peaks of Ni 2p_1/2_ and 2p_3/2_ of Ni_2_Co_1_S_4_ appeared at 873.3 and 855.7 eV, and the peaks of Co 2p_1/2_ and 2p_3/2_ of Ni_2_Co_1_S_4_ appeared at 796.6 and 780.7 eV. For comparison, the peaks of Ni 2p_1/2_ and 2p_3/2_ of Ni_2_Co_1_S_4_-E appeared at 872.9 and 855.3 eV, and the peaks of Co 2p_1/2_ and 2p_3/2_ of Ni_2_Co_1_S_4_-E appeared at 796.2 and 779.6 eV. The binding energies of Ni and Co after alkali etching decreased obviously, indicating the change of chemical environment of the elements of Ni and Co. Combined with XRD results, the decrease of binding energy of Ni and Co may be attributed to the decomposition of the Ni_2_Co_1_S_4_ after the alkali etching. From Figure 2f, the signal of S in Ni_2_Co_1_S_4_-E disappeared, suggesting that the decomposition of the Ni_2_Co_1_S_4_ once again. From Figure 2c,e, the peaks of Ni 2p_1/2_ and 2p_3/2_ of Ni_2_Co_1_Al LDH appeared at 872.8 and 855.3 eV, and the peaks of Co 2p_1/2_ and 2p_3/2_ of Ni_2_Co_1_Al LDH appeared at 796.1 and 780.1 eV. For comparison, the peaks of Ni 2p_1/2_ and 2p_3/2_ of Ni_2_Co_1_Al LDH-E appeared at 872.9 and 855.3 eV, and the peaks of Co 2p_1/2_ and 2p_3/2_ of Ni_2_Co_1_Al LDH-E appeared at 795.6 and 779.8 eV. The binding energy of Ni is almost unchanged, while the binding energy of Co shifts significantly, indicating that Co is more easily affected by alkali etching. From Figure 2g, the signals of Al decreased obviously, indicating the leaching of Al during the alkali etching process. From Figure 2h, the fitted peak at 529.24 eV in Ni_2_Co_1_S_4_-E is assigned to the M-O (Ni-O or Co-O) bond, which should be ascribed to the Ni(OH)_2_ and CoOOH produced by the decomposition of Ni_2_Co_1_S_4_. From Figure 2i, the sharp declining of adsorbed oxygen peak intensity of Ni_2_Co_1_Al LDH-E at 532.53 eV should be ascribed to the desorption of adsorbed oxygen after alkali etching.

From Appendix A, the Ni_2_Co_1_Al LDH and Ni_2_Co_1_Al LDH-E exhibit a typical morphology of hexagonal nanosheet, which are consistent with the morphology of LDH in previous reports [23]. Clear and bright diffraction spots can be found in Appendix A. These diffraction spots form two diffraction circles which should be ascribed to the (113) and (012) planes of Ni_2_Co_1_Al LDH. As shown in Appendix A, the elements of Ni, Co, Al, and O are equably dispersed on the surface of Ni_2_Co_1_Al LDH, indicating that the preparation process is successful. As shown in Appendix A, the diffraction spots become blurred, suggesting that the crystallinity of Ni_2_Co_1_Al LDH is decreased slightly after alkali etching. Comparing the EDS mappings of Ni_2_Co_1_Al LDH and Ni_2_Co_1_Al LDH-E, similar element contents indicate that alkali etching has little effect on the components of Ni_2_Co_1_Al LDH. As shown in Figure 3a, the Ni_2_Co_1_S_4_ obtained by sulfidation exhibits a morphology of an amorphous nanosheet, indicating that the sulfidation process destroys the stable hexagonal structure. The 0.2315 nm lattice spacing is assigned to the (400) plane of Ni_2_Co_1_S_4_ (JCPDS No.73-1704). As shown in Figure 3c, the Ni_2_Co_1_S_4_-E displays a morphology of nanosheets, which is different from the morphology before alkali etching, indicating that the sample of Ni_2_Co_1_S_4_-E have been decomposed completely. Combined with the XRD result, the 0.2429 and 0.2660 nm lattice spacings are assigned to the (101) plane of CoOOH (JCPDS No. 07-0169) and (101) plane of Ni(OH)_2_·0.75H_2_O (JCPDS No. 38-0715). From Figure 3b, the feeble signal of Al indicates that the Al has not been completely removed from the lattice of the Ni_2_Co_1_Al LDH, the feeble signal of S, and the intense signal of O indicate that the element of S is replaced by OH^−^ gradually during alkali etching.

To obtain the microstructure information of the as-obtained materials, the BET measurement of Ni_2_Co_1_S_4_, Ni_2_Co_1_S_4_-E, Ni_2_Co_1_Al LDH, and Ni_2_Co_1_Al LDH-E is carried out, the corresponding N_2_ adsorption/desorption isotherms and pore-size distribution (inset) curves are shown in Figure 4 and the relevant data for specific surface area, pore volume, and average diameter of these samples are shown in Appendix A. The specific surface areas of Ni_2_Co_1_S_4_, Ni_2_Co_1_S_4_-E, Ni_2_Co_1_Al LDH, and Ni_2_Co_1_Al LDH-E are 7.1648, 31.0860, 58.8592, and 63.2865 m^2^ g^−1^, respectively. The pore volumes of Ni_2_Co_1_S_4_, Ni_2_Co_1_S_4_-E, Ni_2_Co_1_Al LDH, and Ni_2_Co_1_Al LDH-E are 0.040893, 0.151196, 0.376361, and 0.376361 cm^3^ g^−1^, respectively. The average pore diameters of Ni_2_Co_1_S_4_, Ni_2_Co_1_S_4_-E, Ni_2_Co_1_Al LDH, and Ni_2_Co_1_Al LDH-E are 18.4100, 14.9734, 13.6747, and 13.0023 m^2^ g^−1^, respectively. The Ni_2_Co_1_Al LDH-E exhibits a maximal specific surface area due to the unique layer structure of LDH and pore-forming by alkali etching. The slight decrease in pore diameter and average pore volume of Ni_2_Co_1_Al LDH-E when compared with Ni_2_Co_1_Al LDH should be ascribed to the small size of the newly formed pores by alkali etching which dragged down the average data. The specific surface area of Ni_2_Co_1_S_4_ is lower than Ni_2_Co_1_Al LDH obviously due to the destruction of layered structure by sulfidation. After the alkali etching of Ni_2_Co_1_S_4_ is carried out, the Ni_2_Co_1_S_4_-E is obtained and exhibits a higher specific surface area than initial Ni_2_Co_1_S_4_ due to the decomposition from sulfide to hydroxide. These results are consistent with those of XRD, XPS, FTIR, and TEM, indicating that the Ni_2_Co_1_S_4_ possesses dissatisfied alkali stability.

### 2.2. Electrochemical Properties

To evaluate the electrochemical performance of the as-prepared Ni_2_Co_1_Al LDH, Ni_1_Co_2_Al LDH, Ni_2_Co_1_S_4_, and Ni_1_Co_2_S_4_, the CV curves at a scan rate of 30 mV s^−1^ and GCD curves at a current density of 1 A g^−1^ are carried out, as shown in Figure 5a,b. The integral areas of Ni_2_Co_1_S_4_ and Ni_1_Co_2_S_4_ from CV curves are higher than those of Ni_2_Co_1_Al LDH, Ni_1_Co_2_Al LDH, and the discharge times of Ni_2_Co_1_S_4_ and Ni_1_Co_2_S_4_ from GCD curves are longer than those of Ni_2_Co_1_Al LDH and Ni_1_Co_2_Al LDH, indicating that sulfidation possesses a superior enhancing effect on the electrochemical properties of NiCoAl LDH. The specific capacitances of Ni_2_Co_1_Al LDH, Ni_1_Co_2_Al LDH, Ni_2_Co_1_S_4_, and Ni_1_Co_2_S_4_ at 1 A g^−1^ are 271.72, 343.37, 718.51, and 1312.87 F g^−1^, respectively, and the specific capacities of Ni_2_Co_1_Al LDH, Ni_1_Co_2_Al LDH, Ni_2_Co_1_S_4_, and Ni_1_Co_2_S_4_ are 37.06, 47.17, 98.46, and 181.10 mAh g^−1^, respectively. From Figure 5c, the rate capabilities of Ni_2_Co_1_Al LDH, Ni_1_Co_2_Al LDH, Ni_2_Co_1_S_4_, and Ni_1_Co_2_S_4_ from 1 to 7 A g^−1^ are 70.41, 32.15, 80.28, and 52.96%, respectively. The charge transfer resistances (*R*_ct_) of Ni_2_Co_1_Al LDH, Ni_1_Co_2_Al LDH, Ni_2_Co_1_S_4_, and Ni_1_Co_2_S_4_ fitted by the equivalent circuit diagram (Figure 5i) are 2.13, 17.00, 3.81, and 5.93 Ω, respectively (Figure 5d). Ni_2_Co_1_S_4_ possesses superior electrical conductivity, outstanding specific capacitance and satisfied rate capability. Although Ni_2_Co_1_Al LDH exhibits a superior alkali stability, the specific capacitance of Ni_2_Co_1_Al LDH is lower than that of two sulfides (Ni_2_Co_1_S_4_ and Ni_1_Co_2_S_4_) obviously. Hence, Ni_2_Co_1_Al LDH does not meet the standard of advanced electrode materials. Based on the above, the Ni_2_Co_1_S_4_ is selected as the target electrode material for subsequent tests. The CV curves of Ni_2_Co_1_S_4_ at various scan rates are shown in Figure 5e. The strong redox peaks should be ascribed to the Ni^2+^/N^i3+^ and Co^2+^/Co^3+^ peaks. The possible electrochemical reaction is shown as follows:(1)$${\mathrm{Ni}}_{2}{\mathrm{Co}}_{1}S_{4}{+2\mathrm{OH}}^{-}↔{\mathrm{CoS}}_{x}{\mathrm{OH}+2\mathrm{Ni}}_{2}S_{4-x}{\mathrm{OH}+2e}^{-}$$

Additionally, the oxidation and reduction peaks of CV curves shift to the direction of high potential and low potential, respectively, without any shape change, indicating a considerable cycling stability. The capacitance retention of Ni_2_Co_1_S_4_ from 1 to 30 A g^−1^ is 20.74%, exhibiting glorious rate capability (Figure 5f). From Figure 5g, Ni_2_Co_1_S_4_ exhibits superior cycling stability of 85.48% capacitance retention after 5000 cycles. In addition, the shapes of GCD curves of Ni_2_Co_1_S_4_ before and after cycling exhibit a stable shape. In addition, the discharge time of the Ni_2_Co_1_S_4_ at 5000th cycle decreases slightly when compared with the discharge time at the 1st cycle, suggesting an excellent cycling stability. According to Figure 5h, the *R*_ct_ values before and after cycling are 3.815 and 18.53 Ω, respectively, suggesting that the decrease of specific capacitance of Ni_2_Co_1_S_4_ is mainly due to the decrease of electrical conductivity.

To evaluate the influence of alkali etching on the electrochemical performance of electrode material, the electrochemical tests of as-etched samples are carried out, as shown in Appendix A. The specific capacitances of Ni_2_Co_1_S_4_-E, Ni_1_Co_2_S_4_-E, Ni_2_Co_1_Al LDH-E, and Ni_1_Co_2_Al LDH-E at 1 A g^−1^ are 612.13, 482.97, 118.47, and 109.35 F g^−1^, respectively, all of which are much lower than that of electrode materials before etching, indicating that the obvious negative effect of alkali etching on the electrochemical performance. Ni_2_Co_1_S_4_ possesses poor alkali stability. However, the cycling stability of Ni_2_Co_1_S_4_ is excellent and is not affected by prolonged alkali immersion. This result may be ascribed that the alkali concentration on the surface of the electrode material is insufficient during the process of the electrochemical test.

To analyze the electrochemical behavior of Ni_2_Co_1_S_4_, the CV curves with small scan rates are carried out, as shown in Figure 6a. The peak current (*I*_p_, A) and scan rate (*v*, mv s^−1^) obey the relationship as follows [24]:(2)$$\log I_{p}=b\log v+\log a$$
where *a* and *b* are the adjustable parameters. The *b* value is concerned with the charge storage mechanism: (1) *b* = 0.5, represents the diffusion-controlled mechanism, (2) *b* ≥ 1.0, represents the surface capacitance-controlled mechanism, (3) 0.5 < *b* < 1.0, represents a mixing capacitance mechanism. According to Figure 6b, the *b*-value of Ni_2_Co_1_S_4_ from anodic and cathodic peaks are 0.66937 and 0.77223, respectively, suggesting that both surface capacitance effect and diffusion-controlled capacitance dominate the electrochemical reaction. The detailed capacitance contribution ratio can be calculated from the following formula [25]:(3)$$I/v^{0.5}=k_{1}v^{0.5}+k_{2}$$
where *I* (A) represents the current at the fixed potential, and *k*_1_ and *k*_2_ are the constants. *k*_1_*v* and *k*_2_*v*^0.5^ represent the charge stored mechanism by the surface capacitance effect and diffusion-controlled capacitance effect. The CV curves and fitted integral area at 1.1, 1.4, and 1.7 mV s^−1^ are shown in Figure 6c–e and display 87.02, 83.09, and 80.18% diffusion-controlled capacitance effect, respectively. From Figure 6f, the contribution proportion of diffusion-controlled capacitance effect increases with the increase of scan rates. The result should be ascribed that high current density leads to the shrinkable diffusion time and diffusion distance.

To evaluate the practical application potential, the commercial active carbon (AC) is used as the negative material. The specific capacitances of AC at 1 A g^−1^ are 150 F g^−1^ (Appendix A). Figure 7a and b exhibit the CV and GCD curves of the Ni_2_Co_1_S_4_//AC ASC device at various potential windows. The CV curves exhibit a quasi-quadrilateral shape within the potential range of 1.0–1.6 V. However, when the potential window reached 1.8 V, obvious polarizations at the high potential region appeared. Similarly, a distinct charge platform appeared in the GCD curves with the potential window of 1.6 V. Hence, the above results indicate that the optimal potential window of ASC device is 1.6 V. The CV curves of ASC device at various scan rates exhibit a stable and homomorphic shape, indicating that the ASC device possesses satisfied stability (Figure 7c). The specific capacitances of ASC device at 1, 2, 3, 4, 5, and 6 A g^−1^ are 54.625, 31.5, 23.81, 19.625, 16.25, and 13.39 F g^−1^, respectively (Figure 7d), the corresponding rate capability from 1 to 6 A g^−1^ is 20.54 % (Figure 7e). The ASC device exhibits a satisfied cycling stability of 61.29% capacitance retention after 2000 cycles (Figure 7f), and the corresponding *R*_ct_ value increases from 29.52 to 135 Ω (Figure 7g), indicating that the decrease of specific capacitance of ASC device is mainly due to the decrease of electrical conductivity. Compared with the three-electrode system, the cycling stability decreases obviously. The results should be ascribed to the negative influence of the negative material of commercial AC. The ASC device displays an appreciable energy density of 18.7 Wh kg^−1^ at 800 W kg^−^^1^ (Figure 7h), which is higher than that of previously reported Ni/Co-based ASC devices [26,27,28,29,30,31]. In addition, as shown in Figure 7i, the LED indicator can be lit up by two ASC device in series, exhibiting a desirable application potential.

## 3. Discussion

In summary, CoAl LDH exhibits a poor alkali stability, and NiAl LDH exhibits an outstanding alkali stability. XRD and TEM analyses prove that the alkali stability of CoAl LDH is improved effectively by doping of Ni. The obtained Ni_1_Co_2_Al LDH and Ni_2_Co_1_Al LDH exhibit a superior alkali stability, but the corresponding specific capacitances are insufficient. Considering the positive influence of the electrical conductivity, the derived Ni_1_Co_2_S_4_ and Ni_2_Co_1_S_4_ from Ni_1_Co_2_Al LDH and Ni_2_Co1Al LDH are also prepared by one-pot sulfidation. The excellent electrical conductivity endows the two sulfides with excellent electrochemical properties. Typically, Ni_2_Co_1_S_4_ exhibits superior specific capacitance of 1312.87 F g^−1^ (181.10 mA h g^−1^) at 1 A g^−1^, preeminent rate capacity of 20.74 % capacitance retention at 30 A g^−1^, and splendid cycling stability of 85.48% capacitance retention after 5000 cycles. Ni_2_Co_1_S_4_ displays more better electrochemical performance than that of Ni_1_Co_2_S_4_. However, the alkali stability of Ni_2_Co_1_S_4_ is worse than that of Ni_1_Co_2_S_4_. After alkali stability testing, Ni_2_Co_1_S_4_ decomposed to Ni(OH)_2_ and CoOOH, while Ni_1_Co_2_S_4_ barely decomposed. Combined with XPS analysis, we speculate that the variable valence state of Co and the solubility of Al in alkali solution are the fundamental reasons for the poor alkali stability of CoAl LDH and Ni_2_Co_1_S_4_. In addition, due to the influence of solubility product constant, sulfide has an inevitable tendency to transform to hydroxide in a high concentration of KOH. Despite its poor alkali stability, Ni_2_Co_1_S_4_ showed excellent cycling stability in the three-electrode system. The result may be ascribed that the alkali concentration on the surface of the electrode material is insufficient during the process of the electrochemical test. Interestingly, although alkali etching does not destroy the crystalline phase of the materials with better alkali stability, such as Ni_2_Co_1_Al LDH and Ni_1_Co_2_S_4_, it also leads to an impaired electrochemical performance. The phenomenon needs to be further studied to understand the possible reaction mechanism. In a word, there is no direct correlation between alkali stability and electrochemical properties. Alkali etching has a negative effect on the electrochemical properties of the materials.

## 4. Materials and Methods

### 4.1. Electrode Materials Preparation

Ni_1_Co_2_Al LDH was prepared via one-pot hydrothermal method. Briefly, 1 mmol Ni(NO_3_)·6H_2_O, 2 mmol Co(NO_3_)·6H_2_O, 1 mmol Al(NO_3_)_3_·9H_2_O and 6 mmol urea were added into the 50 mL breaker with 35 mL deionized water and stirred for 10 min. The obtained mixture was transferred to a Teflon-lined stainless-steel autoclave and maintained at 120 °C for 12 h. After that, the obtained sample was washed centrifugally by deionized water at least 5 times and dried at 60 °C for 12 h. The preparation method of Ni_2_Co_1_Al LDH is similar to Ni_1_Co_2_Al LDH except for using 2 mmol Ni(NO_3_)·6H_2_O and 1 mmol Co(NO_3_)·6H_2_O as raw materials. Ni_1_Co_2_S_4_ was prepared from Ni_1_Co_2_Al LDH by sulfidation. Briefly, 30 mg Ni_1_Co_2_Al LDH and 6 mmol Na_2_S·6H_2_O were added into the 50 mL breaker with 35 mL deionized water and stirred for 10 min. The obtained mixture was transferred to a Teflon-lined stainless-steel autoclave and maintained at 120 °C for 4 h. After that, the obtained sample was washed centrifugally by deionized water at least 5 times and dried at 60 °C for 12 h. The preparation method of Ni_2_Co_1_S_4_ is similar to Ni_1_Co_2_S_4_ except for using Ni_2_Co_1_Al LDH as precursor. In addition, 200 mg Ni_1_Co_2_Al LDH, Ni_2_Co_1_Al_1_ LDH, Ni_1_Co_2_S_4_, or Ni_2_Co_1_S_4_ was added into a 250 mL breaker with 200 mL 3 M KOH solution and stirred for 24 h. The obtained materials are named as Ni_1_Co_2_Al LDH-E, Ni_2_Co_1_Al_1_ LDH-E, Ni_1_Co_2_S_4_-E, or Ni_2_Co_1_S_4_-E, respectively. For comparison, NiAl LDH, CoAl LDH, NiAl LDH-E, and CoAl LDH-E were prepared.

### 4.2. Materials Characterization

The crystalline phase of the as-prepared materials was confirmed by X-ray diffractometer (XRD, XD-2) from 10–80° with a scan rate of 4° min^−1^. The apparent morphology and distribution of elements were obtained by transmission electron microscope (TEM, FEI Tecnai G2 F20 X-Twin) with an energy dispersive X-ray spectrometer (EDS). The specific surface area was collected by surface area analyzer (BET, Gemini VII 2390). Functional groups and molecular structure were analyzed by the Fourier transform infrared spectrometer (FTIR, Cary 630) from 4000–400 cm^−1^. The element kinds, contents, and valence states were determined by X photoelectron spectroscopy (XPS, ThermoFisher ESCALAB Xi+, Waltham, MA, USA) from 0–1350 eV.

### 4.3. Electrochemical Measurements

The cyclic voltammetry (CV), galvanostatic charging/discharging (GCD) measurements, and electrochemical impedance spectroscopy (EIS) were measured using 3 M KOH as an electrolyte on electrochemical workstation (Chenhua CHI760E) in a three-electrode system with the Pt plate (1.5 × 1.5 cm^2^) as a counter electrode, the Ni foam coated by electrode materials as a working electrode, and an Hg/HgO electrode as the reference electrode. The working electrode was prepared using the electrode materials, carbon black, and polyvinylidenefluoride (PVDF) with a mass ratio of 7:2:1. The specific capacitance (*C*_m_, F g^−1^) and specific capacity (*Q*_m_, mAh g^−1^)) are calculated based on the following equations:(4)$$C_{m}=\frac{2I\times \int_{0}^{\Delta t}Vdt}{m\times \Delta V^{2}}$$
(5)$$Q_{m}=\frac{\int_{t_{1}}^{t_{2}}Idt}{3.6m}$$
where *I*, *V*, Δ*t*, and *m* are the discharge current (*A*), potential window (*V*), discharge time (s), and mass of active material (g). The potential window has been considered the voltage drop.

An asymmetric supercapacitor (ASC) device was fabricated using Ni_2_Co_1_S_4_ as the positive material and commercial active carbon as the negative material. The specific capacitance (*C*, F g^−1^), energy density (*E*, Wh kg^−1^), and power density (*P*, W kg^−1^) are calculated based on the following equations:(6)$$C=\frac{I}{m}\int_{t_{1}}^{t_{2}}\frac{1}{Vt}dt$$
(7)$$E=\frac{C\Delta V^{2}}{7.2}$$
(8)$$P=\frac{3600E}{\Delta t}$$
where *m* (g) is the total mass of positive and negative materials. The mass ratio of positive and negative materials was calculated by the following formula:(9)$$\frac{m^{+}}{m^{-}}=\frac{C^{-}\times \Delta V^{-}}{C^{+}\times \Delta V^{+}}$$
wherein *m^+^* (g), *C*^+^ (F g^−1^), and Δ*V*^+^ (V) are the mass, specific capacitance, potential window of the positive materials; *m*^−^ (g), *C*^−^ (F g^−1^), and Δ*V*^−^ (V) are the mass, specific capacitance, and potential window of the negative materials.

## 5. Conclusions

In this work, the alkali stability and electrochemical performance of a series of Al-based LDH and corresponding derivate Ni/Co-based sulfides are analyzed clearly. The alkali stability of CoAl LDH is so poor and can be improved effectively by doping of Ni. Ni_2_Co_1_Al LDH and Ni_1_Co_2_Al LDH exhibit a satisfied alkali stability with an unsatisfied electrochemical performance. The electrochemical performance of Ni_2_Co_1_S_4_ and Ni_1_Co_2_S_4_ is better than that of Ni_2_Co_1_Al LDH and Ni_1_Co_2_Al LDH due to the superior electrochemical conductivity. The alkali stability of Ni_2_Co_1_S_4_ is worse than that of Ni_1_Co_2_S_4_. After alkali stability testing, Ni_2_Co_1_S_4_ decomposed to Ni(OH)_2_ and CoOOH, while Ni_1_Co_2_S_4_ barely decomposed. The reason may be ascribed that the variable valence state of Co and the solubility of Al in alkali solution are the fundamental reasons for the poor alkali stability of CoAl LDH and Ni_2_Co_1_S_4_. Ni_2_Co_1_S_4_ displays more better electrochemical performance than that of Ni_1_Co_2_S_4_. Despite its poor alkali stability, Ni_2_Co_1_S_4_ showed excellent cycling stability of 85.48% capacitance retention after 5000 cycles in the three-electrode system. The results may be ascribed that the alkali concentration on the surface of the electrode material is insufficient during the process of a electrochemical test. In a word, there is no direct correlation between alkali stability and electrochemical properties. Alkali etching has a negative effect on the electrochemical properties of the electrode materials.

## Data Availability

Not applicable.

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
