# Peer review of "Atomic Scale Optimization Strategy of Al-Based Layered Double Hydroxide for Alkali Stability and Supercapacitors"

_ijms, 2022, doi:10.3390/ijms231911645_

Round 1
Reviewer 1 Report
The manuscript tries to describe the preparation of a series of Al-based layered double hydroxide (LDH), and the study of the materials' corresponding alkali stability and electrochemical performance. The title and abstract are appropriate for the content of the text. The experiments were well conducted, and the analysis was well performed. However, the manuscript is disorganized (poorly written).
Comments
· The first paragraph in the introduction (lines 29-37) should be omitted.
· Section 4 (Materials and Methods) should come next to the introduction.
· Acronyms such as LDH (line 16), XRD (line 93), XPS (line 142), EDS (line 181), and FTIR (line 139) need to be defined for first-time use.
Author Response
The manuscript tries to describe the preparation of a series of Al-based layered double hydroxide (LDH), and the study of the materials' corresponding alkali stability and electrochemical performance. The title and abstract are appropriate for the content of the text. The experiments were well conducted, and the analysis was well performed. However, the manuscript is disorganized (poorly written).
Response: Thanks for your evaluation of our manuscript. According to your comments, the manuscript has been carefully revised. Please check out the details in revised manuscript.
- The first paragraph in the introduction (lines 29-37) should be omitted.
Response: We apologize for not checking it carefully before submitting. The first paragraph in the introduction has been deleted.
- Section 4 (Materials and Methods) should come next to the introduction.
Response: Thanks for your careful work. The section of Materials and Methods has been moved to the end of the Introduction.
- Acronyms such as LDH (line 16), XRD (line 93), XPS (line 142), EDS (line 181), and FTIR (line 139) need to be defined for first-time use.
Response: Thanks for your careful work. The full names of all abbreviations have been added when they are used for the first time.

Reviewer 2 Report
The authors prepared several Al-based hydroxide and used them as the electrodes in supercapacitors. The electrochemical performances and the stability in alkali electrolyte were investigated. The research was reasonably designed, and the materials were systematically characterized.
However, there are several things that can be improved.
Delete the first paragraph in the Introduction section.
The caption of Figure 4 is wrong.
The full names of all abbreviations should be shown when they are used for the first time.
The captions of Figure 7 is not clear.
The introduction was not well organized or comprehensive, and didn’t cover the recent significant updates in the research field.
The authors focused on four samples, Ni1Co2Al, Ni2Co1Al, Ni1Co2S4, Ni2Co1S4, and on their stability in alkali electrolyte. Therefore, the chemical compositions, crystal structures, and electrochemical performances of those samples before and after cycling were needed to be compared to draw any conclusion on stability. However, the electrochemical performances were place in the Supplementary Information, and not compared to those before cycling systematically. The presentation of the results need to be reorganized.
The sulfide samples weighed a lot in the Results and Discussion. However, as the authors said, Ni2Co1Al also exhibited good stability. Why Ni2Co1Al was not characterized as much as Ni2Co1S4? The reason of the emphasis should be clearly stated.
The reasoning about the alkali stability of Ni2Co1Al was not rigor. According to the evidence before and after etching in Figure S5, the crystal structure changed, and the changes in element contents calculated on EDS mapping were not negligible. The evidence supported that Ni2Co1Al was not very stable in alkali solution, which is the opposite of the conclusion the authors stated on page 6. It is confusing.
Overall, there are flaws in the research design and the scientific presentation. The results were not strong enough to support the conclusions in the manuscript. I would say reconsider this manuscript after Major Revisions.
Author Response
The authors prepared several Al-based hydroxides and used them as the electrodes in supercapacitors. The electrochemical performances and the stability in alkali electrolyte were investigated. The research was reasonably designed, and the materials were systematically characterized. However, there are several things that can be improved.
Response: Thanks for your evaluation of our manuscript. According to your comments, the manuscript has been carefully revised. Please check out the details in revised manuscript.
- Delete the first paragraph in the Introduction section.
Response: We apologize for not checking it carefully before submitting. The first paragraph in the introduction has been deleted.
- The caption of Figure 4 is wrong.
Response: Thanks for your careful work. The caption of Figure 4 is revised.
Figure 4. N2 adsorption/desorption isotherms and pore-size distribution (inset) of (a) Ni2Co1S4, (b) Ni2Co1S4-E, (c) Ni2Co1Al LDH and (d) Ni2Co1Al LDH-E.
- The full names of all abbreviations should be shown when they are used for the first time.
Response: Thanks for your careful work. Acronyms such as LDH, XRD, XPS, EDS, and FTIR have been defined when they are used for the first time.
- The captions of Figure 7 is not clear.
Response: Thanks for your careful work. The caption of Figure 7 is revised appropriately.
Figure 7. The performance evaluation of ASC device. (a) CV curves at 30 mV s-1 with different potential windows, (b) GCD curves at 3 A g-1 with different current densities, (c) CV curves at different scan rates, (d) GCD curves at different current densities, (e) rate capacities, (f) cycling stability and the GCD curves before and after cycling (inset), (g) Nyquist plots before and after cycling, (h) the Ragone plot of energy and power densities, (i) the LED indicator lit up by two ASC devices.
- The introduction was not well organized or comprehensive, and didn’t cover the recent significant updates in the research field.
Response: Thanks for your careful work. The introduction has been revised, and recent updates has been clarified.
- The authors focused on four samples, Ni1Co2Al, Ni2Co1Al, Ni1Co2S4, Ni2Co1S4, and on their stability in alkali electrolyte. Therefore, the chemical compositions, crystal structures, and electrochemical performances of those samples before and after cycling were needed to be compared to draw any conclusion on stability. However, the electrochemical performances were place in the Supplementary Information, and not compared to those before cycling systematically. The presentation of the results needs to be reorganized.
Response: Thanks for your careful work. We also believe that, it is very important for this research to analyze the morphology and crystal phase changes before and after cycling. The XRD and SEM were conducted and shown in Fig. R1 and R2 respectively. Unfortunately, the results cannot provide any useful information due to the tiny loading mass of Ni2Co1Al LDH and Ni2Co1S4 on Ni foam and the strong interference of Ni foam. From the XRD pattern, the peaks at around 6.73o, 21.43o, 23.82o, 37.70o, 44.38o, 51.68o, 64.25o, 76.19o and 77.38o are the characteristic peak of Ni foam. The characteristic peaks of Ni2Co1Al LDH and Ni2Co1S4 are not found. Similarity, the SEM image cannot distinguish the sample morphology after cycling due to the Interference of additive of carbon black. Furthermore, electrochemical performance is the most important property to evaluate the application prospect of electrode materials. The electrochemical performance of these materials is found in the order as Ni1Co2Al LDH < Ni2Co1Al LDH < Ni1Co2S4 < Ni2Co1S4. Hence, the Ni2Co1S4 was selected as the main analysis object due to its optimal electrochemical performance.
Fig. R1. XRD patterns of Ni2Co1Al LDH and Ni2Co1S4 before and after cycling and Ni foam.
Fig. R2. SEM images of Ni2Co1Al LDH and Ni2Co1S4 before and after cycling.
- The sulfide samples weighed a lot in the Results and Discussion. However, as the authors said, Ni2Co1Al also exhibited good stability. Why Ni2Co1Al was not characterized as much as Ni2Co1S4? The reason of the emphasis should be clearly stated.
Response: Thanks for your careful work. As an electrode material in supercapacitor, the electrochemical performance of the electrode material is more important than the alkali stability. We hope to obtain advanced electrode materials with outstanding alkali stability by reasonable modification methods. Ni2Co1Al LDH exhibits a superior alkali stability. However, the specific capacitance of Ni2Co1Al LDH is lower than that of two sulfides (Ni2Co1S4 and Ni1Co2S4). Hence, Ni2Co1Al LDH does not meet the standard of advanced electrode materials. The relevant statement has been added in the revised manuscript.
Although Ni2Co1Al LDH exhibits a superior alkali stability. However, the specific capacitance of Ni2Co1Al LDH is lower than that of two sulfides (Ni2Co1S4 and Ni1Co2S4) obviously. Hence, Ni2Co1Al LDH does not meet the standard of advanced electrode materials. Based on above, the Ni2Co1S4 is selected as the target electrode material for subsequent tests.
- The reasoning about the alkali stability of Ni2Co1Al was not rigor. According to the evidence before and after etching in Figure S5, the crystal structure changed, and the changes in element contents calculated on EDS mapping were not negligible. The evidence supported that Ni2Co1Al was not very stable in alkali solution, which is the opposite of the conclusion the authors stated on page 6. It is confusing.
Response: Thanks for your careful work. XRD is the most direct evidence to evaluate the alkali stability of electrode materials. After alkali etching, obvious diffraction peaks of NiAl LDH, Ni1Co2Al LDH and Ni2Co1Al LDH for (003) and (006) planes are observed at around 11.3o and 23.1o (Fig. R3a), indicating that the three materials possess outstanding alkali stability. However, the characteristic peaks of CoAl LDH for (003) and (006) planes are disappeared after alkali etching, indicating the CoAl LDH exhibits an inferior alkali stability. In addition, Ni1Co2Al LDH-E exhibits a characteristic peak of CoOOH at 19.8o, indicating that a small amount of Ni1Co2Al LDH-E decomposes into CoOOH. Hence, the alkali stability of these samples are found in the order as NiAl LDH≈Ni2Co1Al LDH>Ni1Co2Al LDH.
As shown in Fig. R3b and e, TEM diffraction spots of Ni2Co1Al LDH-E blurred. The reason should be ascribed to the decrease of crystallinity caused by alkali etching. The changes in TEM diffraction spots cannot be used as direct evidence to evaluate the alkali stability. Fig. R4a and b can further illustrate the effect of alkali etching on crystal crystallinity. After alkali etching, the peak intensity of (003) plane for NiAl LDH and Ni2Co1Al LDH are decreased obviously, indicating an obvious destructive effect of alkali etching for crystal crystallinity. Besides, due to the ex-situ analysis, the content change from EDS mapping can only be used as a reference to evaluate its alkali stability.
Fig. R3. The XRD patterns of (a) NiAl LDH, (b) CoAl LDH, (c) Ni2Co1Al LDH and Ni1Co2Al LDH before and after alkali etching. The TEM images of Ni2Co1Al LDH before (d) and (e) after alkali etching.
Fig. R4. The change of peak intensities of (a) NiAl LDH and (b) Ni2Co1Al LDH before and after alkali etching.
- Overall, there are flaws in the research design and the scientific presentation. The results were not strong enough to support the conclusions in the manuscript. I would say reconsider this manuscript after Major Revisions.
Response: Thanks for your careful reviewing and pertinent advice.

Round 2
Reviewer 2 Report
The questions from reviewers have been addressed. With the corrections and the modifications, the quality of the manuscript is much improved. I agree to publish this manuscript.